# Do conspiracy theories efficiently signal coalition membership? An experimental test using the *"Who Said What?"* design

**Mathilde Mus**[1,2]*, **Alexander Bor**[1], **Michael Bang Petersen**[1]

**1** Department of Political Science, Aarhus University, Aarhus, Denmark, **2** Département d'études Cognitives, École Normale Supérieure—PSL, Paris, France

* mathilde.mus@ens.psl.eu

## Abstract

Theoretical work in evolutionary psychology have proposed that conspiracy theories may serve a coalitional function. Specifically, fringe and offensive statements such as conspiracy theories are expected to send a highly credible signal of coalition membership by clearly distinguishing the speaker's group from other groups. A key implication of this theory is that cognitive systems designed for alliance detection should intuitively interpret the endorsement of conspiracy theories as coalitional cues. To our knowledge, no previous studies have empirically investigated this claim. Taking the domain of environmental policy as our case, we examine the hypothesis that beliefs framed in a conspiratorial manner act as more efficient coalitional markers of environmental position than similar but non-conspiratorial beliefs. To test this prediction, quota sampled American participants (total N = 2462) completed two pre-registered *Who-Said-What* experiments where we measured if participants spontaneously categorize targets based on their environmental position, and if this categorization process is enhanced by the use of a conspiratorial frame. We find firm evidence that participants categorize by environmental position, but no evidence that the use of conspiratorial statements increases categorization strength and thus serves a coalitional function.

## Introduction

Why do people believe and share conspiracy theories? Three psychological motives have been put forward by previous research [1, 2]: a) epistemic motives, referring to people's need to understand and navigate their environment [3]; b) existential motives, relating to people's need to feel secure and in control of their environment [4]; c) social motives, by which people can manage their reputation and signal their membership to a coalition [5, 6]. For example, the conspiratorial belief that global warming is a hoax can at the same time provide an explanation for temperatures that may be perceived as incongruent with global warming (e.g., colder winters), prevent an existential anguish over the impending climate catastrophe, and signal an engagement with environmental-skeptic groups. In this paper, we focus on the proposed social motives associated with the endorsement of conspiratorial beliefs. Specifically, we explore the claim that endorsing conspiracy theories can send a credible signal of coalition membership, a claim which to our knowledge has not yet been empirically evaluated.

**Data Availability Statement:** All informational and reproducibility materials are available at https://osf.io/2ufhy/.

**Funding:** This research was funded by grant no. CF18-1108 (ROPH: Center for Research on Online

Political Hostility) from the Carlsberg Foundation to MBP. The funders had no role in study design, data collection and analysis, decision to publish, or preparation of the manuscript. https://www.carlsbergfondet.dk/en.

**Competing interests:** The authors have declared that no competing interests exist.

From an evolutionary perspective, the coalitional function of beliefs arises from the fact that beliefs can serve as a cue to distinguish ingroup members from outgroup members as they, for example, signal familiarity with cultural norms and customs. For most of their evolutionary history, humans lived in small hunter-gatherer groups where both coordination with ingroup members and group-based defense against outgroups acted as strong selection pressures. As a consequence, the human mind has evolved a series of specialized mechanisms for coalitional management to respond to these adaptive challenges [7, 8]. The first crucial step in coalitional management is the detection of alliances, namely being able to detect who is likely to belong to one's ingroup or to one's outgroup prior to an interaction. This requires specialized cognitive adaptations capable of making predictive forecasts about coalitional membership on the basis of the available cues [9, 10]. Empirical evidence in favor of such an "*alliance detection system*" in the human mind, keeping track of relevant coalitional cues, has been established [10–12]. Such cues could be physical in nature, taking the form of clothing and ornaments for example, but could also be contained in shared attitudes. Indeed, as people who share beliefs, values and opinions tend to cooperate and form alliances, it is likely that the mind evolved to perceive cues of shared attitudes as coalitional markers. In line with this, prior research has shown that political attitudes are encoded as coalitional markers by the *alliance detection system* [12].

However, there should be variability in the degree to which various shared beliefs act as coalitional cues. The best coalitional signals should be the ones that clearly indicate loyalty to one group and differentiation from other groups [13]. Beliefs that undermine the person's ability to join other coalitions, by triggering irreversible reputational costs, can thus acquire a strategic advantage [14, 15]. Indeed, the more likely beliefs are to lead to a rejection by other social groups, the more the belief-holder should appear as a loyal member of the group in line with these beliefs. This phenomenon of earning credibility by reducing one's options–here by reducing one's chances to join other coalitions–has been called a "*burning bridges*" strategy [16]. Burning bridges is a classical strategy in game theory where removing or limiting a player's options can paradoxically improve payoffs [17]. This commitment device signals loyalty towards a targeted group by greatly limiting cooperation opportunities with other groups.

In the context of burning bridges with other coalitions by endorsing certain beliefs, how can this strategy be best achieved? One possibility is to use *fringe beliefs*, that is statements that contradict common sense or established facts and that are held by a minority of people. Indeed, they should act as an honest signal of coalition membership both because of the specialized knowledge they convey [5] and the resulting rejection expected from most other groups. Another efficient way to make belief statements burn bridges is by being offensive towards other coalitions, attacking either their intentions or their competence [16], which is also very likely to result in rejection by targeted groups.

## Conspiracies and environmental policy

In this manuscript, we have chosen to focus on a current and specific example of the burning bridges strategy: the endorsement of conspiracy theories. The present manuscript seeks to empirically test the hypothesis that endorsements of conspiratorial beliefs efficiently act as coalitional markers through bridge-burning. A conspiracy theory is commonly defined as the belief that a group of agents secretly acts together with malevolent intent [18, 19]. Most conspiracy theories are thus inherently offensive: they accuse some actors of harming innocent people, either directly (as in the chemtrail conspiracy) or indirectly by concealing relevant information and "covering up tracks". Another common case is that conspiracy theories deny grievances or important achievements of certain actors (e.g. Holocaust deniers or the *9/11*

*Truth Movement*; moon-landing hoax), thereby also fostering inter-group conflict. Moreover, many conspiracy beliefs oppose mainstream narratives and are often held by small minorities (e.g. Reptilian conspiracies), thereby also possessing a fringe element. Endorsing fringe beliefs accusing other groups of malevolent intent is therefore a costly behavior because of the expected ostracization the belief-holder faces. For instance, Redditors active in conspiracy communities get moderated and receive negative replies more often than users who are never active in conspiracy communities [20]. Conspiracy believers themselves appear to be aware of these costs: those who share conspiracy theories believe that others evaluate them negatively and expect to face social exclusion [21]. These findings indicate that if one is seeking to signal their loyalty by alienating other groups, endorsing conspiracies may be a potential successful strategy.

As our case, we have chosen to focus on conspiratorial beliefs related to the environment. Environmental conspiracy theories have been rising over the last decades, especially those regarding climate change denial [22, 23]. These beliefs may hinder the implementation of effective policies urgently required to mitigate global warming [23]. Indeed, conspiracy theories can negatively affect policy making both directly by fostering opposition to evidence-based measures, and indirectly by diverting useful time and resources in order to address them [24].

To test if endorsements of environmental conspiracy theories act as more efficient coalitional markers than non-conspiratorial environmental beliefs, we study activation patterns of the *alliance detection system*. The alliance detection system must be able to pick up on which alliance categories are currently shaping people's behavior and inhibit non-relevant alliance categories. Indeed, alliances may change, and people always belong to more than one coalition [11]. Therefore, presenting new alliance categories relevant to a current situation should both increase categorization along the new dimensions and decrease categorization by other alliance categories that do not act as good predictors of alliance relationships at the moment. Experimental evidence has shown that, although race is a strong alliance cue in contemporary American society, the alliance detection system readily downregulates categorization by race when more relevant alliance categories–such as basketball team membership, charity membership or political support–are presented [10–12]. Thus, categorization by race is an ideal indicator to determine if a presented cue acts as a coalitional marker.

We extend previous research by positing that the framing of beliefs has an impact on the strength of categorization processes. In line with the *burning bridges* account, we hypothesize that beliefs with a conspiratorial dimension send a more credible signal of coalition membership than beliefs without a conspiratorial dimension. Consequently, we expect conspiratorial statements to increase categorization by the relevant alliance category and to decrease categorization by the non-relevant alliance categories. In the context of the present research, environmental position acts as the relevant alliance category and race as the non-relevant one. We therefore hypothesize that endorsements of environmental beliefs framed in a conspiratorial manner should, compared to similar but non-conspiratorial beliefs, lead to an increase in categorization by environmental position and a decrease in race categorization.

## Materials and methods

The three present studies bear on how people categorize speakers on the basis of race and environmental position in the presence or absence of conspiratorial arguments. In all studies, implicit social categorization was measured using the "*Who Said What*?" memory confusion paradigm, following standards in the literature [10–12]. Data for this project has been collected in full compliance with the law of the Danish National Committee on Health Research Ethics

(§14.2), which specifies that survey and interview studies that do not include human biological materials are exempted from an ethical approval by the committee. All surveys started with a written informed consent form.

## Procedure

The *Who-Said-What* experimental paradigm proceeds in three stages. Following the procedure of Petersen [25, 26], we used a shortened version of the task adapted for web surveys of representative samples.

**a. Presentation phase.** When entering the study, participants were told that they would be viewing a discussion about the environment among pro-environmental individuals and environmental skeptics. After providing written informed consent to take part in the experiment, participants then watched a sequence of eight pictures of young men in their 20's, each paired with a statement about the environment displayed for 20 seconds. Participants were simply asked to form an impression of the target individuals by looking at the pictures and reading the statements.

**b. Distractor task.** A distractor task was then used to reduce rehearsal and recency effects. In this task, participants were asked to list as many countries as they could in one minute.

**c. Surprise recall phase.** In the surprise recall phase, each statement was presented in a random order and participants were asked to choose which of the eight simultaneously displayed targets had uttered the given statement. The errors made in the recall phase reveal whether the mind spontaneously categorizes targets along a dimension. If so, targets belonging to the same category along this dimension are more likely to be confused with each other than targets from different categories.

Finally, participants answered a few demographic questions and were thanked.

## Materials and general design

Four statements expressed the view that more should be done to protect the environment ("pro-environmental") and four that less should be done ("environmental-skeptic"). To ensure ecological validity, all statements were modelled after real views expressed on social media sites in debates about the environment. The presentation order of the statements was randomized within the constraint that they should alternate between a pro-environmental position and an environmental-skeptic position to create a discussion frame. Each statement first expressed an environmental position identical between the control condition and the treatment condition, and then provided a justification whose framing–conspiratorial or non-conspiratorial–differed across conditions. The control condition can be considered as a placebo rather than a "pure control" in which the treatment is absent [27]. Indeed, to the extent possible, we sought to design similar justifications between the two conditions, which varied solely by the presence or absence of a conspiratorial dimension in order to maximize experimental control. Fig 1 presents a set of sample statements in both conditions. The full list of statements can be found in the S1 File.

Statements in the control condition were somewhat shorter than statements in the treatment condition. However, even if the additional length leads to more errors, there is no reason to expect a bias towards either more within-category or between-category errors.

Following standard practice for Internet-based experiments in psychology [29, 30], materials were pre-tested. 100 American participants (32 women; mean age = 36.9 years) were recruited on Amazon Mechanical Turk and compensated with pay. The pre-test assessed that the statements respected the study's criteria: (a) statements designed to be "pro-environmental" were rated as significantly more pro-environmental (M = 5.43, SD = 1.49) than

|  | Non-conspiratorial (control) | Conspiratorial (treatment) |
|---|---|---|
| **Pro-environmental** | « The Department of Agriculture should ban GMOs.<br><br>*GMOs have a negative impact on the environment as well as on our health.* » | « The Department of Agriculture should ban GMOs.<br><br>*For decades, agribusinesses have suppressed data showing that GMOs harm the environment and our health.* » |
| **Environmental-skeptic** | « The government needs to stimulate economic growth.<br><br>*Degrowth, as promoted by environmentalists, would harm our way of life.* » | « The government needs to stimulate economic growth.<br><br>*The environmentalists promoting degrowth are being hired by foreign states to harm our way of life.* » |

**Fig 1. Illustration of experimental stimuli.** Each statement is composed of two sentences. The first one, here written in ordinary type, corresponds to an environmental position and is identical across conditions. The second sentence, here written in italics, corresponds to a justification of the environmental position that varies across conditions, being either framed as conspiratorial (treatment condition) or not (control condition). Statements were paired with target photos taken from the Center for Vital Longevity Face Database [28].

"environmental-skeptic" statements (M = 3.48, SD = 2.16); F(1, 788) = 218.38), p < .001; (b) statements designed to be "conspiratorial" were rated as significantly more conspiratorial (M = 5.35, SD = 1.53) than "non-conspiratorial" statements (M = 4.36, SD = 1.96); F(1, 783) = 61.98, p < .001. Complete information about the pre-test can be found in the S2 File.

Four speakers were white and four were black in order to induce race categorization. Men targets were used because previous studies showed that race categorization for men targets is more resistant to change than race categorization for women targets [10–12], creating a more stringent test for our hypothesis. Target photos were taken from the Center for Vital Longevity Face Database [28]. Speakers' race was balanced across the environmental dimension, such that both environmental positions were defended by two white and two black targets, removing the correlation between race and present alliances. Within this constraint, the pairing between targets and statements was randomized.

## Measures

Each answer in the surprise recall task is categorized as either a correct answer, a within-category error or a between-category error. A within-category error is made when the chosen target belongs to the same category as the correct response. For example, a within-category error for race is made when a statement uttered by a black target is wrongly attributed to one of the other black targets. In a between-category error, the two confused responses belong to different categories. Because a target cannot be confused with itself (as that would be a correct answer), within-category errors are less frequent than between-category errors. To correct for this asymmetry in base-rates, the number of between-category errors for both race and environmental position is multiplied by 0.75 for each participant [31]. Finally, a categorization score is calculated as the difference between these two types of errors. A mean categorization score significantly above zero signals that participants spontaneously categorize targets along the given dimension, namely race or environmental position in the present study. One-sample t-tests are run in order to determine if categorization scores, both for race and environmental position, are positive. Two-sample t-tests are run to examine whether there is an increase in

environmental position categorization and a decrease in race categorization when statements are framed in a conspiratorial manner rather than in a non-conspiratorial manner. Following standards in the literature [10–12], categorization scores are translated into a measure of effect size, Pearson's r, with higher values corresponding to stronger categorization along a given dimension. All pre-registered directional hypotheses are tested with one-tailed tests.

## Pilot study

To our knowledge, environmental position has never been tested as a coalitional cue in the Who-Said-What paradigm. Before studying differences in activation patterns of the *alliance detection system* in relation to the conspiracy variable, a pilot study was run in order to test if the mind spontaneously categorizes people according to their views on environmental policy.

### Participants

Based on effect sizes found by Pietraszewski et al. (2015) [12], a power analysis indicated that a sample of 100 persons would allow to detect a small-sized effect with a probability of 90% using a two-tailed t-test. 120 American participants were recruited from the online platform Amazon Turk (46 women; mean age = 35.9 years) and were paid 1$ to complete the experiment.

### Design

This study included only one experimental condition in which all statements were presented in their non-conspiratorial form (i.e. the control condition in Fig 1). Indeed, this pilot study solely aimed at establishing the existence of categorization by environmental position using the *Who-Said-What* experimental paradigm.

### Results and discussion

Categorization scores were significantly above zero for both race (r = .35, p < .001, 95% CI [0.20, 0.48]) and environmental position (r = .24, p < .001, 95% CI [0.05, 0.39]). We thus first replicate the finding that the mind spontaneously encodes race as an alliance category [10–12]. This result can be related to the central place of race in American politics, where persisting racial divisions, resentments, and group loyalties have been evidenced [32]. The results also demonstrate that, in parallel to race categorization, the mind spontaneously categorizes people according to their views on environmental policy. This result is in line with the findings of Pietraszewski and colleagues (2015) [12] who found evidence in favor of categorization by political attitudes.

This result also implies that environmental policy positions offer a good case to study the effects of conspiratorial framing on categorization strength. Indeed, non-conspiratorial environmental beliefs constitute a good baseline for our hypothesis which predicts a decline of race categorization and an increase in environmental categorization, as initial high levels of categorization by race and moderate levels of categorization by environmental position were obtained.

## Study 1

Study 1 explores the specified hypothesis that environmental beliefs framed in a conspiratorial manner should act as efficient coalitional cues and thus lead to stronger categorization by environmental position and weaker categorization by race than similar but non-conspiratorial beliefs. The study design and analysis plan were pre-registered at OSF https://osf.io/6aumy. In

our pre-registered studies, we planned to exclude participants who failed attention checks. However, because attention checks were implemented post-treatment, these exclusions could bias our causal estimates [33]. Accordingly, we deviate from our pre-registrations and include all respondents in the analyses reported below. In the S3 File, we report pre-registered analyses on attentive respondents yielding identical substantive conclusions.

## Participants

A power analysis using a one-tailed test and 5% alpha level indicated that to detect a small effect size (d = .2) in a two-samples t-test with 90% power, 858 participants are required. 1200 U.S. citizens were recruited from the online platform Lucid. Lucid uses quota sampling to ensure that the sample margins resemble population distributions in terms age, gender, race, education, and region. Lucid provides samples consisting of more diverse and less experienced participants than those recruited on Amazon Mechanical Turk. This platform has been validated as a good alternative online panel marketplace [34]. Only participants who finished the survey were included in the analysis, leaving 1147 participants (554 women; mean age = 43.1 years).

## Design

There were two between-subjects conditions in this study: in the control condition, all statements were presented in their non-conspiratorial form whereas in the treatment condition, all statements were conspiratorial (cf. Fig 1). Participants were randomly assigned to one of the two conditions.

## Results and discussion

In both the control and the treatment condition, participants categorized target speakers on the basis of environmental position (control: r = .15, p < .001, 95% CI [0.07, 0.23]; treatment: r = .10, p = .01, 95% CI [0.01, 0.19]) and race (control: r = .43, p < .001, 95% CI [0.36, 0.49]; treatment: r = .34, p < .001, 95% CI [0.26, 0.41]). These results replicate the findings of the pilot study, highlighting their robustness. As predicted, categorization by race was significantly lower in the treatment condition compared to the control condition (t = 1.83, df = 1005, p = .03). Categorization by environmental position, however, was not significantly larger in the treatment condition (t = 0.60, df = 964, p = .72; see Fig 2). If anything, categorization by environmental position was weaker when conspiratorial justifications were offered.

The findings therefore support only one of the specified predictions. When statements were framed in a conspiratorial manner rather than in a non-conspiratorial manner, there was a significant decrease in race categorization but not a significant increase in categorization by environmental position.

To further explore our results, we performed additional analyses to investigate whether the predicted results may be conditioned by the direction of the statements (pro-environmental or environmental-skeptic) or by participants' political worldviews. Indeed, it has been shown that people selectively apply their conspiracy thinking in line with their political identity [35]. Regarding climate-related conspiracies, Uscinski and Olivella (2017) [36] review evidence that "Republicans are more likely to believe that climate change is a hoax while Democrats are more likely to believe that oil companies are hiding solutions to climate change" (p.2). However, we do not find that the direction of the statements in our study moderates the effect of conspiratorial framing on environmental categorization scores (p = 0.75). We also do not find evidence that participants' political ideology or level of environmental concern moderates the studied relationship (p = 0.11 and p = 0.48 respectively). Hence, participants do not appear to

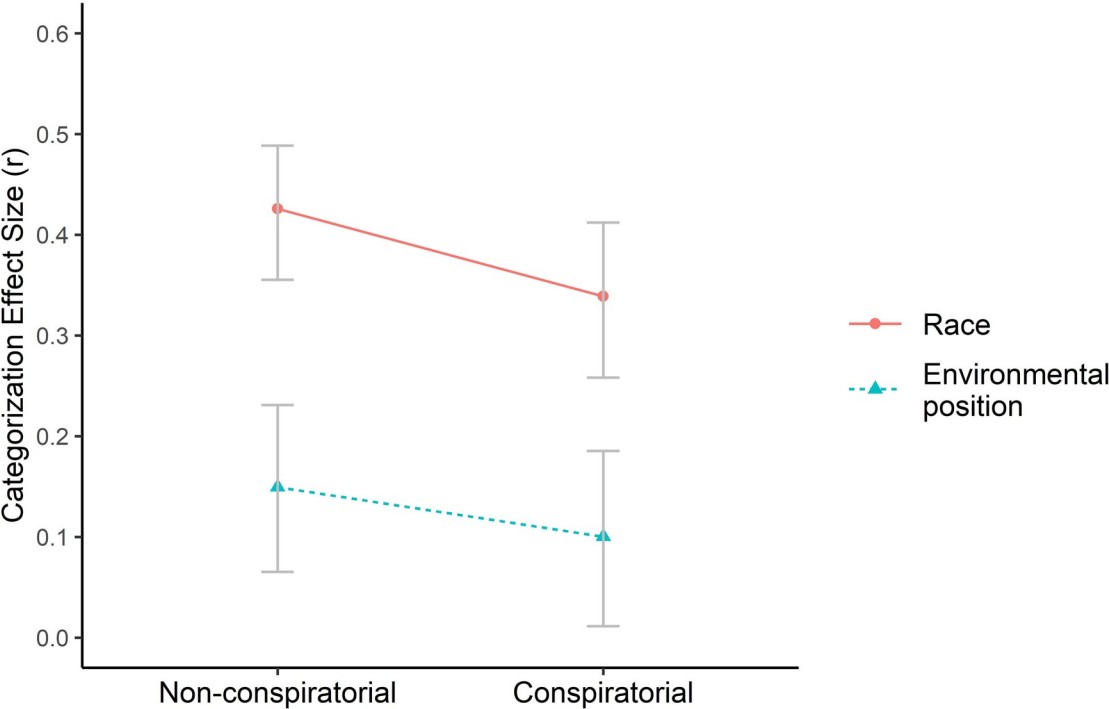

**Fig 2. Categorization by race and environmental position when statements are framed either in a non-conspiratorial (control) or conspiratorial (treatment) form (N = 1147).** Only race categorization was significantly lowered by the use of a conspiratorial frame. The reported numbers are effect sizes (r). Error bars correspond to bootstrapped 95% confidence intervals.

categorize targets differently according to either the direction of the statements or their political worldviews.

A possible confound influencing the results of this study is that conspiratorial justifications could serve as an indicator of affiliation with an independent coalition composed of all conspiracy theorists. In the psychological literature, it has indeed been argued that conspiracy theorists possess a specific "conspiratorial mindset" displaying for example a low level of interpersonal trust. People believing in one conspiracy theory tend to believe in other conspiracies [37] even if they contradict each other or are entirely fictitious [38]. Conspiratorial statements may also signal low competence, a dimension that the mind automatically encodes [39]. In both cases, conspiracy theorists might therefore be categorized as belonging to the same coalition. If this was true, we would still expect a reduction in categorization by race because a novel coalition cue was introduced (conspiratorial arguments), but unlike our original prediction, categorization by environmental position would either be reduced or remain unaffected because the novel cue blurs differences between the two positions. Indeed, whereas in the control group the two opposing environmental views are contrasted, in the treatment group both views could be seen as branches of the same coalition (i.e., of conspiracy theorists).

## Study 2

Study 2 was designed to investigate further the unexpected results of Study 1, by eliminating the potential confound that conspiratorial justifications may serve as an indicator of affiliation with an independent coalition composed of all conspiracy theorists. To do so, we modify the treatment condition by eliminating half of the conspiratorial frames compared to Study 1. As our focus remains on the potential use of conspiratorial sentences to boost categorization

across another coalitional dimension, we do *not* create a new conspiracy dimension orthogonal to race and environmental position. Instead, we align the conspiratorial dimension with environmental position such that either all four pro-environmental statements are conspiratorial and no environmental-skeptic statements are, or vice versa. We then test whether conspiratorial arguments strengthen categorization by environmental position if only one side uses them.

If this is true, we expect categorization by environmental position to increase in the treatment group compared to the control group, as all conspiracy theorists now share the same environmental stance. Furthermore, if indeed conspiratorial asymmetries boost environmental position as a *coalitional* cue, we expect categorization by race to decrease in the treatment group. Similarly to Study 1, categorization along a dimension is measured as the propensity to make more errors between targets who share this dimension (e.g. race or environmental position) than between targets who differ regarding this dimension. The study design and analysis plan were pre-registered at OSF https://osf.io/43trw.

## Participants

1200 American participants were recruited from the online platform Lucid. Only participants who finished the survey were included in the analysis, leaving 1195 participants (610 women; mean age = 45.1 years).

## Design

As in Study 1, there were two between-subjects conditions to which participants were randomly assigned. The only difference in design between Study 1 and Study 2 lies in the treatment condition. In the treatment condition of Study 2, only half of the statements were conspiratorial and the conspiracy dimension was superposed with environmental position: either all four pro-environmental statements were conspiratorial and no environmental skeptic-statements were, or vice versa.

## Results and discussion

In both the control and the treatment conditions, participants categorized target speakers on the basis of environmental position (control: r = .15, p < .001, 95% CI [0.07, 0.23]; treatment: r = .21, p < .001, 95% CI [0.13, 0.29]) and race (control: r = .46, p < .001, 95% CI [0.40, 0.51]; treatment: r = .45, p < .001, 95% CI [0.38, 0.51]). However, neither of our hypotheses were supported by the data. Despite a slight increase in categorization by environmental position, the change does not reach significance at conventional levels (t = 1.22, df = 1193, p = .11, see Fig 3). Neither was categorization by race significantly decreased in the treatment condition compared to the control condition (t = 0.04, df = 1193, p = .51).

Hence, the findings of Study 2 do not support the prediction that conspiratorial frames boost categorization by environmental position when only one side uses them, as only a weak effect in the expected direction was found.

## Discussion

Several lines of theory within evolutionary psychology have emphasized the social function that false and extreme beliefs could serve, in line with the *burning bridges* account [5, 16]. Taking as our case environmental conspiracy beliefs, we empirically investigated the hypothesis that conspiracy theories act as efficient coalitional markers. However, the reported experiments do not provide significant evidence in favor of this hypothesis.

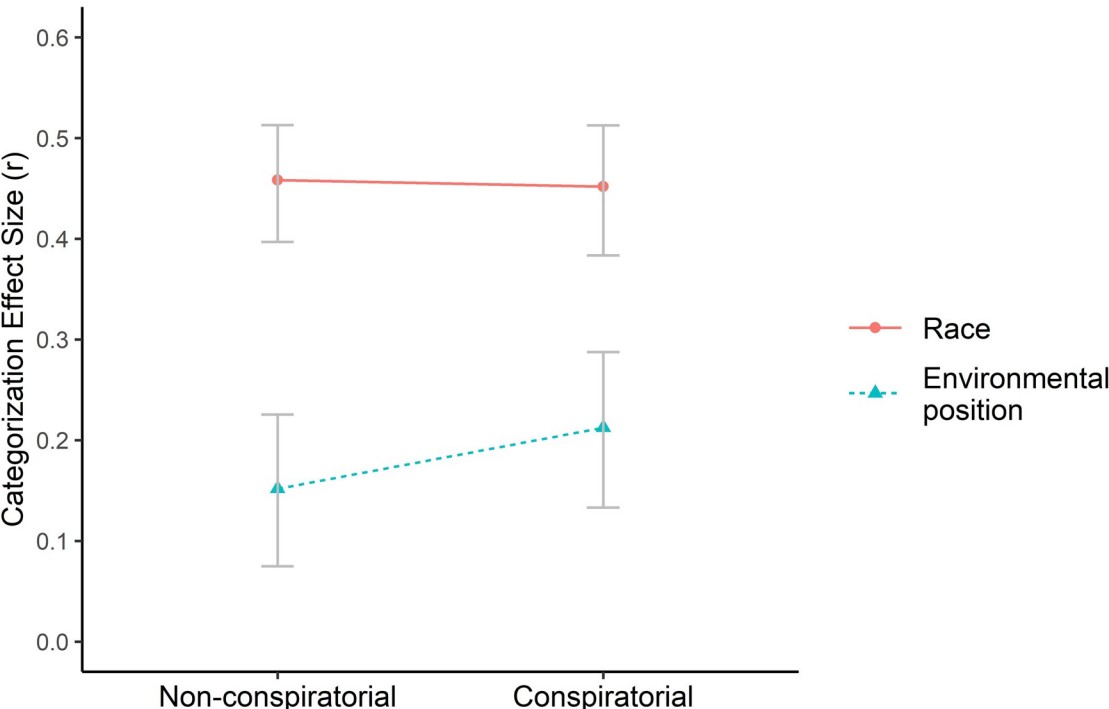

**Fig 3. Categorization by race and environmental position when statements are either framed in a non-conspiratorial form (control) or in a conspiratorial form aligned with environmental position (treatment) (N = 1195).** Both race environmental position categorization scores do not significantly differ across conditions. The reported numbers are effect sizes (r). Error bars correspond to bootstrapped 95% confidence intervals.

As a first step, we demonstrated that environmental position elicits categorization in the *Who-Said-What* design. This result is consistent with the findings of Pietraszewski and colleagues (2015) [12] who established that political positions act as coalitional markers. Our main studies then tested whether, when the environmental position was justified with a conspiracy theory, categorization by environmental position increased and categorization by race–another potential but here irrelevant alliance dimension–decreased.

Study 1 found evidence only for one of the specified predictions: race categorization significantly decreased when environmental statements were framed in a conspiratorial manner instead of a non-conspiratorial manner, but categorization by environmental position did not significantly increase. Study 2 was designed to eliminate a confound that could influence Study 1's results, namely that conspiratorial justifications may serve as an indicator of affiliation with an independent coalition composed of all conspiracy theorists. However, Study 2 only found a weak effect in favor of the coalitional cue conveyed by conspiracy theories when removing this confound.

Therefore, the reported experiments do not provide strong evidence in favor of the hypothesis that conspiratorial beliefs act as coalitional markers, beyond the political position they indicate. These results may first suggest that the "burning-bridges" component of beliefs sends a weaker coalitional signal than what has been theoretically suggested in the literature [5, 6]. They may also suggest that the coalitional function of conspiratorial beliefs more generally plays a smaller explanatory role than the other motivations identified as drivers of conspiracy theories such as epistemic motives and existential concerns [1, 2]. For instance, when conspiracy theories are endorsed in online contexts where anonymity is the rule, it is likely that the

belief-holder will be less affected by reputational costs than in offline contexts. In this case, the coalitional motivation of sharing such content may become weaker.

However, our findings might also reflect false negative results due to chance, as well as methodological artefacts. Indeed, although we conducted two rigorously designed and high-powered experiments on diverse online samples, our studies suffer from some limitations. A first possible explanation for null results obtained in our studies may lie in our choice of the studied alliance category. As no empirical research on the *alliance detection system* has previously used environmental position as an alliance category, it is possible that this domain behaves differently from other alliance categories. For instance, environmental position might still not be perceived as a category divisive enough in the population and hence would activate less strongly the alliance detection system than other more classic political coalitions. In our sample, there were around four times more participants who believed that federal spending should be increased to protect the environment rather than decreased, whereas the proportion of participants identifying as Democrats versus Republicans were similar. Despite the acknowledged correlation between political orientation and environmental concern [40], the latter proved to be less divisive than political orientation in our sample. Future research may seek to replicate our experiments with more divisive and commonly used alliance categories, such as partisanship or broader political ideology.

A second methodological point that future work could address is related to the difference in the perceived conspiratorial dimension between treatment and control statements. When designing the statements, there was a trade-off between maximizing this conspiracy gap across conditions and preserving experimental control. Our ambition in designing the sentences had been two-fold: 1) maximizing ecological validity by rooting sentences in environmental statements found online and 2) maximizing internal validity by ensuring high similarity between treatment and control sentences. Although a pre-test assessed that conspiracy ratings significantly differed between treatment and control statements (mean difference = 0.99, SD = 0.12, d = .56), it is possible that the gap between the two conditions was not large enough to elicit the predicted results. Future research could therefore investigate the generalizability of our findings by modifying our sentence stimuli. A first approach could be to design statements based on more theoretical considerations (e.g., varying in their *burning bridges* components), which could both make the manipulations stronger and help to dissect the various mechanisms involved in alliance signaling. Another more ecological approach would be to use extreme examples of conspiratorial beliefs (e.g., Pizzagate, Reptilian conspiracies) for proof-of-concept purposes, despite the loss of experimental control occurring when comparing the effects with non-conspiratorial statements.

In conclusion, we tested the hypothesis that endorsements of conspiracy theories are processed as coalitional cues, using environmental conspiracy theories as our case. We did not find clear empirical support for this hypothesis, using a series of *Who-Said-What* experiments. As this study is, to our knowledge, the first to empirically test the coalitional function of conspiracy theories, future research could attempt to replicate the reported experiments while addressing the potential methodological limitations underlined above, in order to further explore the validity of the evolutionary framework under scrutiny.

## Supporting information

**S1 File. Full list of statements.**
(PDF)

**S2 File. Pre-test of statements.**
(PDF)

**S3 File. Attention checks and analyses without inattentive respondents.**
(PDF)

## Acknowledgments

We are grateful to the members of the Research on Online Political Hostility (ROPH) group at Aarhus University for their insightful comments. We also thank the anonymous referees for their helpful suggestions to improve the manuscript.

## Author Contributions

**Conceptualization:** Mathilde Mus, Alexander Bor, Michael Bang Petersen.

**Data curation:** Mathilde Mus.

**Formal analysis:** Mathilde Mus.

**Funding acquisition:** Michael Bang Petersen.

**Investigation:** Mathilde Mus.

**Methodology:** Mathilde Mus, Alexander Bor, Michael Bang Petersen.

**Project administration:** Mathilde Mus, Michael Bang Petersen.

**Resources:** Michael Bang Petersen.

**Software:** Mathilde Mus, Alexander Bor.

**Supervision:** Alexander Bor, Michael Bang Petersen.

**Visualization:** Mathilde Mus.

**Writing – original draft:** Mathilde Mus.

**Writing – review & editing:** Alexander Bor, Michael Bang Petersen.

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
