## [Decision Letter · Decision Letter 0]

26 Oct 2021

PONE-D-21-29587Do conspiracy theories efficiently signal coalition membership? An experimental test using the “*Who Said What?*” designPLOS ONE

Dear Dr. Mus,

Thank you for submitting your manuscript to PLOS ONE. After careful consideration, we feel that it has merit but does not fully meet PLOS ONE’s publication criteria as it currently stands. Therefore, we invite you to submit a revised version of the manuscript that addresses the points raised during the review process.

We look forward to receiving your revised manuscript.

Kind regards,

Shang E. Ha, Ph.D.

Academic Editor

PLOS ONE

Journal Requirements:

2. Please change "female” or "male" to "woman” or "man" as appropriate, when used as a noun (see for instance https://apastyle.apa.org/style-grammar-guidelines/bias-free-language/gender).

4. Please ensure that you refer to Figure 2 and 3 in your text as, if accepted, production will need this reference to link the reader to the figure.

5. We note that Figure 1 includes an image of a participant in the study.

Reviewers' comments:

Reviewer's Responses to Questions

**Comments to the Author**

1. Is the manuscript technically sound, and do the data support the conclusions?

Reviewer #1: Yes

Reviewer #2: Yes

2. Has the statistical analysis been performed appropriately and rigorously? 

Reviewer #1: Yes

Reviewer #2: Yes

3. Have the authors made all data underlying the findings in their manuscript fully available?

Reviewer #1: Yes

Reviewer #2: Yes

4. Is the manuscript presented in an intelligible fashion and written in standard English?

Reviewer #1: Yes

Reviewer #2: Yes

5. Review Comments to the Author

Reviewer #1: General opinion

- Overall, I think the manuscript does add to our understanding of the evolution of human coalitional psychology, and how conspiratorial beliefs might contribute to it. The hypothesis is plausible, and that the lack of significant results should not be seen as a negative in terms of publication. However there does seem to be a lack of engagement with the conspiracy theory literature, and this harms the integration of the evolution-based hypothesis into that field in the subsequent in-text discussion.

Introduction

- There is some important literature missing regarding conspiratorial beliefs. Specifically, existential concerns are missing (see., Douglas et al, 2017; 2019). This means 1/3 of the established explanations for conspiratorial belief are missing, meaning the background to conspiratorial belief is not covered; it at least requires a mention. Given existential concerns cover fear and security, it seems relevant to coalitional psychology. Specifically, group-level identity (and out-group threat) is categorized under existential concerns as a predictor of belief.

- The example given to illustrate the psychology of conspiracies does not really capture epistemic motives. Most research on that category focuses on aspects such as uncertainty or pattern-recognition, or to explain dramatic or shocking events. Equally, the conspiracy theory regarding Obama’s nationality served initially to delegitimize his presidency in principle and identify him as an outgroup, not to explain policy decisions as “anti-American” or “pro-African” per se. I would find a better example or rephrase the explanation of this conspiracy.

- While I am convinced by the logic that accepting fringe or counter-factual beliefs serves as group-membership cues, the suggestion of “burning bridges” requires more explanation. I would also be keen to see specific empirical examples (rather than book references) of where fringe beliefs lead to ostracization by others, rather that self-exclusion by the believer to signal their commitment to their new community. There is a paper by Van Prooijen and colleagues that is currently under review that does touch on this.

- Overall, I find the premise and the resulting predictions logical, though more engagement with the conspiracy theory literature is needed to truly tie it into an evolutionary framework.

Methods/results

- Regarding the statements/stimuli. I understand the inclusion criteria as per the SI, but judging by the means alone the differences are sometimes minimal even if statistically significant. For an example, if we take a mid-point of 3.5 as neither agree or disagree whether X conspiracy is widely held, 4.23 and 4.5 are not extreme. I would also guess they both are actually significantly above this mid-point. This seems the case for most of the statements, so how likely are they to produce a response in this paradigm. I am not familiar with it, so has this piloting approach - and selection criteria - been shown to be useful in creating stimuli in the past? – this issue is touched on in the discussion, briefly, but an explanation is warranted because it might invalidate the findings as a whole.

- Equally, and this subjective of course, but none seem especially extreme – we live in a world of Qanon and some very identity-charged conspiracies, or ones that are counter-factual boarding on psychopathology (the UK royal family are lizards, for example). So, were not more extreme statements about the environment considered, even if for proof-of-concept purposes?

- Perhaps it is my unfamiliarity with the who-said-what paradigm, but I did have trouble keeping track of exactly what was being measured. I would suggest reminding the reader exactly what categorization means when introducing study 2.

- I would be interested to see whether there was any effect of the direction of the statement as well as the accompanying conspiratorial statement. While there is contention on political ideology in the Conspiracy literature, there to seem to be differences in how individuals on the left and right respond to conspiratorial beliefs that correspond to their general perspective. Has this analysis been performed: i.e., Pro/Anti*control/conspiracy?

Discussion

- The arguments given here for the lack of support for the hypotheses are quite weak. I would like to see the null-results put in the broader context of both evolved coalition psychology and the conspiracy literature before limitations are discussed. Neither literature is given appropriate consideration here. It may suggest that bridge burning is a more nuanced, or a weaker, part of commitment signaling than suggested for example(?).

- I get the impression the study has used conspiracy beliefs as a simple convenient way to probe the coalitional psychology theories. This is fine. If this is the case though, as above, more needs to be said beyond “null hypothesis supported, maybe there were method issues”. If my impression is correct, this would also necessitate a restructuring of the introduction section, with conspiracies simply being a specific and current example of bridge-burning.

- The discussion does mention environmental concerns as a perhaps a less divisive issue, but there is a literature (as mentioned) on left wing Vs right-wing conspiracies that might add to this discussion. It is certainly different when compared to anti-vax beliefs where there is an intersection of left and right. I would recommend the exploratory analysis suggested previously

Reviewer #2: This is an interesting and well-done paper that should be published in PLOS One. I especially the admire the authors' forthrightness in confronting their hypothesis for which they failed to find evidence.

That said, I do have some comments and observations meant to improve the manuscript.

-I was not entirely clear about the relationship between conspiracy belief and bridge burning. It seems to me that I can endorse conspiracy theories without burning bridges; especially in online contexts, the costs of promoting, and then walking away from, various conspiracies seem low. For those unfamiliar with this literature, the authors need to clarify the relationship.

-The authors describe a control condition that would be better described as a placebo. For challenges associated with placebos in survey experiments, consult Velez and Porter (2021). 

-Were the removed participants removed because they followed an attention check pre or post treatment? The authors should clarify. If the attention check occurred post treatment, the authors should re-insert those participants to avoid post-treatment bias.

-The very first paragraph seems to overstate the prevalence of conspiracy beliefs; the claim that "the magnitude and prominence of conspiratorial beliefs is soaring" should either be toned down or tied to a reference that persuasively makes that point.

-There's not nearly enough discussion of the role that racial perceptions may be playing in these studies. Especially as this was administered on U.S. samples, it seems likely to me that participants were judging the stimuli for the race of the person *and only the race* and nothing else. The authors need to elaborate on the relationship between race and the effects observed.

But again, this is well-done and interesting and deserves to be published.

6. PLOS authors have the option to publish the peer review history of their article (what does this mean?). If published, this will include your full peer review and any attached files.

Reviewer #1: No

Reviewer #2: No

---

## [Author Response · Author response to Decision Letter 0]

7 Jan 2022

EDITOR

We have made several corrections to meet PLOS ONE’s style requirements. We have renamed our Figures as Fig 1, Fig 2 and Fig 3 (in the main text, figure captions and file names). We also renamed the supporting information files, which are now submitted as five separate files along the manuscript. 

2. Please change "female” or "male" to "woman” or "man" as appropriate, when used as a noun

We implemented the changes in the revised manuscript.

The full ethics statement now appears in the Methods section.

4. Please ensure that you refer to Figure 2 and 3 in your text as, if accepted, production will need this reference to link the reader to the figure.

We have now added the reference to Figure 2 and 3 in the main text (p.11 and p.13 respectively).

5. We note that Figure 1 includes an image of a participant in the study.

We admit that the source of the images included in Figure 1 was unclear. The images in Figure 1 do not belong to participants of our studies but are taken from the Center for Vital Longevity Face Database, from which eight target pictures were chosen as stimuli. To clarify the source of images in the figure caption, we added “Target photos were taken from the Center for Vital Longevity Face Database [28].” to Fig 1’s caption. 

[Update: After receiving information by the editor that these images must be removed, we removed all pictures from this database and deleted our S3 File which reproduced the target photos used in our experiments. We also added after the relevant reference (28) in the reference list that “Access to the database can be requested at: https://agingmind.utdallas.edu/download-stimuli/face-database/”]

REVIEWER 1

General opinion

- Overall, I think the manuscript does add to our understanding of the evolution of human coalitional psychology, and how conspiratorial beliefs might contribute to it. The hypothesis is plausible, and that the lack of significant results should not be seen as a negative in terms of publication. However there does seem to be a lack of engagement with the conspiracy theory literature, and this harms the integration of the evolution-based hypothesis into that field in the subsequent in-text discussion.

We are grateful for the insightful comments. We agree with the reviewer about the need to incorporate a broader set of references from the conspiracy theory literature, both in the introduction and the discussion. We have made several additions to our manuscript in this regard. These are detailed in the points below.

Introduction

- There is some important literature missing regarding conspiratorial beliefs. Specifically, existential concerns are missing (see., Douglas et al, 2017; 2019). This means 1/3 of the established explanations for conspiratorial belief are missing, meaning the background to conspiratorial belief is not covered; it at least requires a mention. Given existential concerns cover fear and security, it seems relevant to coalitional psychology. Specifically, group-level identity (and out-group threat) is categorized under existential concerns as a predictor of belief.

The introduction now opens with the classification by Douglas et al. (2017; 2019) listing all three explanations for conspiratorial beliefs: 

“Three categories of psychological motives influencing conspiratorial endorsement have been put forward by previous research [1,2]: a) epistemic motives, referring to people’s need to understand their environment, that helps them navigate in it [3], b) existential motives, relating to people’s need to feel secure and in control of their environment [4], c) social motives, by which people can manage their reputation and signal their membership to a coalition [5,6].” (p.2)

- The example given to illustrate the psychology of conspiracies does not really capture epistemic motives. Most research on that category focuses on aspects such as uncertainty or pattern-recognition, or to explain dramatic or shocking events. Equally, the conspiracy theory regarding Obama’s nationality served initially to delegitimize his presidency in principle and identify him as an outgroup, not to explain policy decisions as “anti-American” or “pro-African” per se. I would find a better example or rephrase the explanation of this conspiracy.

We acknowledge the limits of the previous example and introduce a new one that captures all three mentioned motives of conspiracy beliefs (epistemic, existential, social). We are now using the conspiracy of climate change being a hoax, which has the additional advantage of being thematically related to our case study. More specifically, we have added in the introduction the following sentence, after the list of the 3 psychological motives: 

“For example, the conspiratorial belief that global warming is a hoax can at the same time provide an explanation for temperatures that may be perceived as incongruent with global warming (e.g. colder winters), prevent an existential anguish over the impending climate catastrophe, and signal an engagement with environmental-skeptic groups.” (p.2).

- While I am convinced by the logic that accepting fringe or counter-factual beliefs serves as group-membership cues, the suggestion of “burning bridges” requires more explanation. I would also be keen to see specific empirical examples (rather than book references) of where fringe beliefs lead to ostracization by others, rather that self-exclusion by the believer to signal their commitment to their new community. There is a paper by Van Prooijen and colleagues that is currently under review that does touch on this.

We now explain bridge-burning in more detail in the introduction (p.3-4), as this notion was indeed not clear enough.

Regarding the second part of the comment: When introducing conspiracy theories as a specific case of bridge-burning that we wished to investigate (p. 4), we have now added empirical references in which holders of conspiratorial beliefs have been ostracized by others or expect social exclusion for expressing such views (Phadke et al., 2021 ; Lantian et al., 2018). We found these very relevant empirical studies in the review paper by Van Prooijen and colleagues (to be published) mentioned by the reviewer, whom we thank for this reference.

- Overall, I find the premise and the resulting predictions logical, though more engagement with the conspiracy theory literature is needed to truly tie it into an evolutionary framework.

Thank you for this encouraging perspective, we hope that the additions made to the introduction have filled this caveat.

Methods/results

- Regarding the statements/stimuli. I understand the inclusion criteria as per the SI, but judging by the means alone the differences are sometimes minimal even if statistically significant. For an example, if we take a mid-point of 3.5 as neither agree or disagree whether X conspiracy is widely held, 4.23 and 4.5 are not extreme. I would also guess they both are actually significantly above this mid-point. This seems the case for most of the statements, so how likely are they to produce a response in this paradigm. I am not familiar with it, so has this piloting approach - and selection criteria - been shown to be useful in creating stimuli in the past? – this issue is touched on in the discussion, briefly, but an explanation is warranted because it might invalidate the findings as a whole.

The reviewer is right in noticing that differences in characteristics between treatment and control statements are indeed sometimes minimal. This results from a methodological choice of favoring experimental control over a more pronounced difference between the stimuli in the two conditions. We wanted treatment statements to differ from the control statements only in their conspiratorial dimension, to capture the sole effect of “conspiraciness” and not other variables that are likely to vary if the content of the statements was manipulated further. We thus wished the content of the statements to be as similar as possible between the two conditions, except for this conspiratorial dimension. We have tried to make this argument clearer in the Materials and design section (p.7). This methodological choice led us to use conspiratorial statements which are indeed less extreme than they could have been if we had not made this choice. We encourage future research to implement the opposite methodological trade-off: favoring the difference between conditions over experimental control, at least for proof-of-concept purposes as suggested by the reviewer in the next comment. We have made this point more salient in the discussion (p.17).

Regarding the point raised about the mid-point threshold and the fact that some control statements rate above this point, this is also a very relevant comment. In this study, we were interested mainly in a relative phenomenon, namely that beliefs possessing more “burning-bridges” components are more efficient in triggering categorization by environmental position than beliefs that are less prone to burning bridges. This is why we chose the differences in ratings between statements as the relevant statistical test for our pre-test to be validated rather than absolute comparisons with the midpoint. We have added this point in the supplementary file describing the pre-test (S2).

Regarding the piloting approach, while most previous Who-Said-What studies sought to establish whether participants categorize along certain dimensions, here our first ambition was to assess whether conspiratorial statements increase the level of categorization. This necessitated more careful pre-testing of materials: Our goal was not only to offer clear cues on a given category of information, but to manipulate the conspiracy dimension between the control and treatment conditions. Accordingly, we followed standard practice for Internet-based experiments in psychology and pre-tested our materials on a number of dimensions (Reips, 2002; 2007). The goal of the pre-test was to validate that the environmental position conveyed by statements was picked up by participants, and that treatment statements were perceived as more conspiratorial and more likely to burn bridges (i.e. less widely held and more offensive) than control statements. We updated both the manuscript (p.8) and supplementary materials (S2) to better reflect these considerations. 

- Equally, and this subjective of course, but none seem especially extreme – we live in a world of Qanon and some very identity-charged conspiracies, or ones that are counter-factual boarding on psychopathology (the UK royal family are lizards, for example). So, were not more extreme statements about the environment considered, even if for proof-of-concept purposes?

We completely agree with the reviewer on the point raised. We have tried to explain our methodological trade-off in the answer to the previous comment. Future research should indeed use more extreme statements to see if the predicted results are elicited in that case and this is the next step we ourselves will take. We have underlined this direction for future research in the discussion (p.17)

- Perhaps it is my unfamiliarity with the who-said-what paradigm, but I did have trouble keeping track of exactly what was being measured. I would suggest reminding the reader exactly what categorization means when introducing study 2.

We have now reminded the reader of what is being measured when introducing Study 2 and clarified the alternative hypothesis under scrutiny: 

“Study 2 was designed to investigate further the unexpected results of Study 1. It empirically explores the alternative hypothesis that the mind encodes the conspiracy dimension as an alliance category independent from environmental position. This alliance category would be composed of all conspiracy theorists, from both sides of the environmental spectrum. To test this prediction, we align the conspiratorial dimension of statements with environmental position in the treatment condition, such that all conspiratorial statements are either pro-environmental or environmental-skeptic. Similarly to previous studies, categorization along a dimension is measured as the propensity to make more errors between targets that share this dimension (e.g. race, environmental position) than between targets who differ regarding this dimension. If the specified hypothesis is true, categorization by environmental position is expected to increase in the treatment group compared to the control group as all conspiracy theorists share the same environmental stance in this new design, while race categorization should still decrease because a new relevant alliance category is being introduced as in previous studies. ” (p.13-14)

- I would be interested to see whether there was any effect of the direction of the statement as well as the accompanying conspiratorial statement. While there is contention on political ideology in the Conspiracy literature, there to seem to be differences in how individuals on the left and right respond to conspiratorial beliefs that correspond to their general perspective. Has this analysis been performed: i.e., Pro/Anti*control/conspiracy?

This is a very interesting point. We have now performed the analysis pro/anti*control/conspiracy and have found no interaction effect in Study 1 (we have not done the analysis for Study 2 as the environmental dimension of statements is aligned with the conspiracy dimension). We now report the results of this analysis at the end of the Results section of Study 1, as an additional analysis performed to test whether the predicted effect could be conditional on the direction of the statement (pro or anti), citing references that are in line with the phenomenon mentioned by the reviewer. 

Ideally, to test the hypothesis of the reviewer, a three-way interaction poolitical_worldviews*pro/anti*control/conspiracy would be required. However, this statistical test would be underpowered. Thus, we decided to run two-way interactions between political worldviews and our manipulation, to bring additional evidence to the point raised while maintaining acceptable statistical power. In our experiment, we measured participants’ opinion on whether federal spending on environmental protection should be kept the same, increased/decreased a little, increased/decreased moderately or increased/decreased a great deal (7-point scale). We also have access to the political ideology endorsed by participants (Republican or Democrat, 10-point scale) measured by the platform Lucid. We therefore conducted additional analyses in line with the reviewer’s comment, studying the interaction between political worldviews (environmental concern and political ideology) and our experimental manipulation on categorization scores in Study 1. No significant interaction was found either for environmental concern nor political ideology. We have reported these results at the end of the Results section of Study 1 as well: 

“To further explore our results, we performed additional analyses to investigate whether the predicted results may be conditional on the direction of the statements (pro-environmental or environmental-skeptic) and on participants’ political worldviews. Indeed, it has been shown that people selectively apply their conspiracy thinking in line with their political identity (Miller et al., 2016). Regarding climate-related conspiracies, Uscinki and Olivella (2017) review evidence that “Republicans are more likely to believe that climate change is a hoax while Democrats are more likely to believe that oil companies are hiding solutions to climate change” (p.2). However, we do not find that the direction of the statements in our study moderates the effect of conspiratorial framing on categorization scores (p = 0.93). We also do not find evidence that participants’ political ideology or level of environmental concern moderates the studied relationship (p = 0.11 and p = 0.81 respectively).” (p.13)

We are grateful to the reviewer for these very interesting leads to further explore our results.

Discussion

- The arguments given here for the lack of support for the hypotheses are quite weak. I would like to see the null-results put in the broader context of both evolved coalition psychology and the conspiracy literature before limitations are discussed. Neither literature is given appropriate consideration here. It may suggest that bridge burning is a more nuanced, or a weaker, part of commitment signaling than suggested for example(?).

We agree with the reviewer about the missing theoretical implications of our results before limitations are exposed. We have now added a paragraph in the discussion that puts the results into a broader theoretical context (p.16). We have now suggested that either bridge-burning is indeed a weaker part of coalitional signalling than what was thought, or that it is the social function of conspiratorial beliefs more generally that may be weaker than the identified other motivations driving these beliefs (epistemic, existential), especially in online contexts as pointed out by the second reviewer (#1). Limitations are now discussed after this theoretical account.

- I get the impression the study has used conspiracy beliefs as a simple convenient way to probe the coalitional psychology theories. This is fine. If this is the case though, as above, more needs to be said beyond “null hypothesis supported, maybe there were method issues”. If my impression is correct, this would also necessitate a restructuring of the introduction section, with conspiracies simply being a specific and current example of bridge-burning.

We agree about the need to clarify the relationship between conspiracies and bridge-burning, a point also raised by the second reviewer (#1). In the introduction, we have made additions to the section “Conspiracies and environmental policy” to clarify the fact that conspiracies are indeed considered as optimal candidates for bridge-burning and thus should act as efficient coalitional markers: 

“In this paper, we have chosen to focus on a current and specific example of the burning bridges strategy: conspiracy theories. The present manuscript seeks to empirically test the hypothesis that endorsements of conspiratorial beliefs efficiently act as coalitional markers through bridge-burning. A conspiracy theory is commonly defined as the belief that a group of agents secretly acts together with malevolent intent [references] – thus these beliefs are offensive by definition. Moreover conspiracy beliefs oppose mainstream narratives and often held by small minorities, thereby also possessing a fringe element. Endorsing fringe beliefs accusing other groups with malevolent actions is therefore a costly behavior because of the expected ostracization the belief-holder faces.“ (p.4)

In line with what the reviewer suggests in this comment and the previous comment, we have also modified our discussion section by adding theoretical implications of our null results on bridge-burning and the social function of conspiracy theories (explained in the previous answer), to tie into the evolutionary framework outlined in the introduction.

- The discussion does mention environmental concerns as a perhaps a less divisive issue, but there is a literature (as mentioned) on left wing Vs right-wing conspiracies that might add to this discussion. It is certainly different when compared to anti-vax beliefs where there is an intersection of left and right. I would recommend the exploratory analysis suggested previously

As we have found no interaction between the direction of the statement and our experimental manipulation, we have not added this point to the discussion but we detail this relevant hypothesis of a moderation by the direction of the statement when we report this analysis in the manuscript (Results section of Study 1). However, we agree with the reviewer about the need to enrich this point of environmental concern being a potentially less divisive issue in the discussion. We believe it is important to mention the correlation between political orientation and environmental concern because it may go against the argument of environmental concern being a less divisive issue than political orientation (if the correlation is very strong). In our data however, we find evidence of political orientation being more divisive than environmental concern and have added the following sentences in the discussion: 

“For instance, environmental position might still not be perceived as a category divisive enough in the population and hence would activate less strongly the alliance detection system than other more classic political coalitions. In our sample, there were around four times more participants who believed that federal spending should be increased to protect the environment rather than decreased, whereas about the same proportion of participants were favoring the Democrat party or the Republican party (S6). Despite the acknowledged correlation between political orientation and environmental concern [Cruz, 2017], the latter proved to be less divisive than political orientation in our sample. Future research may seek to replicate the reported experiments with more divisive and commonly used alliance categories, such as partisanship or broader political ideology. ” (p.17)

REVIEWER 2

This is an interesting and well-done paper that should be published in PLOS One. I especially the admire the authors' forthrightness in confronting their hypothesis for which they failed to find evidence. That said, I do have some comments and observations meant to improve the manuscript.

We are grateful to the reviewer for this very encouraging perspective and for the time taken to review our manuscript.

1. I was not entirely clear about the relationship between conspiracy belief and bridge burning. It seems to me that I can endorse conspiracy theories without burning bridges; especially in online contexts, the costs of promoting, and then walking away from, various conspiracies seem low. For those unfamiliar with this literature, the authors need to clarify the relationship.

We agree that the relationship between the endorsement of conspiratorial beliefs and bridge-burning was not sufficiently clear, as also pointed out by Reviewer 1. We study conspiracy theories as a current and specific example of the burning-bridges strategy. Conspiratorial beliefs are both fringe and offensive beliefs, two characteristics that are likely to burn bridges with other groups. We have tried to clarify this relationship in the introduction, in the “Conspiracies and environmental policy” section: 

“In this paper, we have chosen to focus on a current and specific example of the burning bridges strategy: conspiracy theories. The present manuscript seeks to empirically test the hypothesis that endorsements of conspiratorial beliefs efficiently act as coalitional markers through bridge-burning. A conspiracy theory is commonly defined as the belief that a group of agents secretly acts together with malevolent intent [references] – thus these beliefs are offensive by definition. Moreover conspiracy beliefs oppose mainstream narratives and often held by small minorities, thereby also possessing a fringe element. Endorsing fringe beliefs accusing other groups with malevolent actions is therefore a costly behavior because of the expected ostracization the belief-holder faces.“ (p.4)

We have also added two empirical references showing that people expressing conspiracy theories are more likely to see their posts or comments on Reddit moderated or receiving negative feedback, and that people expressing conspiracy theories expect to be negatively evaluated, and to be socially excluded by others (p.5). These examples empirically demonstrate that endorsing conspiracy theories tends to burn bridges with outgroups that are not in line with these beliefs.

Finally, we use a commonly used definition of conspiracy theories (“A conspiracy theory is commonly defined as the belief that a group of agents secretly acts together with malevolent intent”) that suggests that these beliefs always burn bridges because of the “malevolent intent” it includes. However, we agree with the reviewer that the extent of bridge-burning and resulting reputational costs very much depend on the context in which these beliefs are endorsed. In an online context where there is anonymity, a conspiracy theorist does not have to face reputational costs, and the coalitional signalling function of endorsing a conspiracy theory may in that case be much weaker. We have added this very interesting point in the discussion (p.16). 

2. The authors describe a control condition that would be better described as a placebo. For challenges associated with placebos in survey experiments, consult Velez and Porter (2021).

When introducing the control condition, we have added the following sentences: 

“The control condition can be considered as a placebo rather than a “pure control” in which the treatment is absent (Porter & Velez, 2021). Indeed, to the extent possible, we sought to design similar justifications between the two conditions, which varied solely by the presence or absence of a conspiratorial dimension”. (p.8)

We have chosen to keep the term “control condition” in the rest of the paper in order not to confuse readers who would not be familiar with the term “placebo” in survey experiments but we hope that this addition allows to raise the point rightfully made. 

We thank the reviewer for mentioning Porter & Velez (2021), who point to the challenge of minimizing researchers’ impact on the creation of placebos. We thought this issue was important to raise so we have also added in the Materials and design section that “to also maximize ecological validity and minimize bias in the design of stimuli, we designed beliefs inspired from environmental statements available on the internet.” (p.8)

3. Were the removed participants removed because they followed an attention check pre or post treatment? The authors should clarify. If the attention check occurred post treatment, the authors should re-insert those participants to avoid post-treatment bias.

This information was indeed missing. The attention check happened post-treatment, which can lead to post-treatment bias (Montgomery et al., 2018). Following the recommendation of the reviewer, we have reinserted those participants and rerun the analyses for our three studies (pilot, study 1, study 2). The results are unchanged and we have reported them in a supplementary file as robustness checks (S5). We have chosen to keep the current results with excluded participants in the main manuscript because we had committed to exclude these participants in the pre-registration of the studies. However, if the reviewer or the editor deem it preferable that the results without excluded participants be presented in the main manuscript rather than in the supplementary information, we will be happy to implement the changes. We have added in the discussion section this caveat of post-treatment bias and the new analyses conducted: 

“However, our findings might also reflect false negative results due to chance, as well as methodological artefacts. Indeed, although we conducted two rigorously designed and high-powered experiments on diverse online samples, our studies suffer from some limitations. As attention checks happened post-treatment, a bias can occur in the exclusion of inattentive respondents (Montgomery et al., 2018). However, when re-running the analyses of all studies without excluding participants, results remain unchanged (S5). ”. (p.17)

4. The very first paragraph seems to overstate the prevalence of conspiracy beliefs; the claim that "the magnitude and prominence of conspiratorial beliefs is soaring" should either be toned down or tied to a reference that persuasively makes that point.

We have removed this claim from the introduction as we have found no reference that persuasively validates this point, we thank the reviewer for raising this issue.

5. There's not nearly enough discussion of the role that racial perceptions may be playing in these studies. Especially as this was administered on U.S. samples, it seems likely to me that participants were judging the stimuli for the race of the person *and only the race* and nothing else. The authors need to elaborate on the relationship between race and the effects observed.

This is an interesting point. Race categorization is indeed the strongest form of categorization taking place in our studies (effects sizes are about twice larger than in the case of categorization by environmental position). However, all our studies find evidence that categorization by environmental position also takes place (as categorization r-scores are significantly above 0). Moreover, multiple studies using the Who-Said-What paradigm show that the mind is able to encode several categories in parallel (Kurzban et al, 2021; Pietrazewski et al., 2014; 2015). But we agree that more emphasis on the large effect of race categorization (stronger than categorization by environmental position) is required, and that this makes sense when working with an American sample. We thus made additions in the part describing the results from the pilot, that focused on proving that both race categorization and categorization by environmental position were taking place: 

“Categorization scores were significantly above zero for both race (r = .40, p < .001, 95% CI [0.24, 0.53]) and environmental position (r = .25, p = .008, 95% CI [0.06, 0.41]). We thus first replicate the finding that the mind spontaneously encodes race as an alliance category [Kurzban et al, 2021; Pietrazewski et al., 2014; 2015]. This result can be related to the central place of race in American politics, where persisting racial divisions, resentments, and group loyalties have been evidenced [Hutchings & Valentino, 2004]. The results also demonstrate that, in parallel to race categorization, the mind spontaneously categorizes people according to their views on environmental policy. “ (p.11)

But again, this is well-done and interesting and deserves to be published.

Thank you!

---

## [Decision Letter · Decision Letter 1]

24 Jan 2022

PONE-D-21-29587R1Do conspiracy theories efficiently signal coalition membership? An experimental test using the “*Who Said What?*” designPLOS ONE

Dear Dr. Mus,

Thank you for submitting your manuscript to PLOS ONE. After careful consideration, we feel that it has merit but does not fully meet PLOS ONE’s publication criteria as it currently stands. Therefore, we invite you to submit a revised version of the manuscript that addresses the points raised during the review process. I was able to get the opinion of the original two reviewers. One felt the paper was ready. The other asked for only minor changes now. Please submit your revised manuscript by Mar 10 2022 11:59PM. If you will need more time than this to complete your revisions, please reply to this message or contact the journal office at plosone@plos.org. Please include the following items when submitting your revised manuscript:A rebuttal letter that responds to each point raised by the academic editor and reviewer(s). You should upload this letter as a separate file labeled 'Response to Reviewers'.A marked-up copy of your manuscript that highlights changes made to the original version. You should upload this as a separate file labeled 'Revised Manuscript with Track Changes'.An unmarked version of your revised paper without tracked changes. You should upload this as a separate file labeled 'Manuscript'.If applicable, we recommend that you deposit your laboratory protocols in protocols.io to enhance the reproducibility of your results. Protocols.io assigns your protocol its own identifier (DOI) so that it can be cited independently in the future. For instructions see: https://journals.plos.org/plosone/s/submission-guidelines#loc-laboratory-protocols. Additionally, PLOS ONE offers an option for publishing peer-reviewed Lab Protocol articles, which describe protocols hosted on protocols.io. Read more information on sharing protocols at https://plos.org/protocols?utm_medium=editorial-email&utm_source=authorletters&utm_campaign=protocols.

We look forward to receiving your revised manuscript.

Kind regards,

Peter Karl Jonason

Academic Editor

PLOS ONE

Journal Requirements:

Reviewers' comments:

Reviewer's Responses to Questions

**Comments to the Author**

1. If the authors have adequately addressed your comments raised in a previous round of review and you feel that this manuscript is now acceptable for publication, you may indicate that here to bypass the “Comments to the Author” section, enter your conflict of interest statement in the “Confidential to Editor” section, and submit your "Accept" recommendation.

Reviewer #1: All comments have been addressed

Reviewer #2: (No Response)

2. Is the manuscript technically sound, and do the data support the conclusions?

Reviewer #1: Yes

Reviewer #2: Yes

3. Has the statistical analysis been performed appropriately and rigorously? 

Reviewer #1: Yes

Reviewer #2: Yes

4. Have the authors made all data underlying the findings in their manuscript fully available?

Reviewer #1: Yes

Reviewer #2: Yes

5. Is the manuscript presented in an intelligible fashion and written in standard English?

Reviewer #1: Yes

Reviewer #2: Yes

6. Review Comments to the Author

Reviewer #1: Having read the manuscript, I feel the authors have addressed the comments raised in the initial review in the new manuscript draft itself and in their response.

Reviewer #2: I applaud the authors for a well-executed revision. The study and its contributions are much more clear. A few remaining points:

1. If I were the authors, I would indeed report results for all subjects, including those who failed the post treatment attention check. The authors acknowledge that these results are what *should* be reported; the results don't change (they say) if those subjects are included; and, perhaps most importantly, this paper is going to be published, and it would be unfortunate if readers focused on this error, rather than the substantive contribution of the paper. In short, I think it's in their interest, and the long-term interests of this paper, to make this change.

2 I would appreciate more details on the modifications made in Study 2. Right now, I don't think I fully grasp how the alignment of "conspiratorial dimension of statements with environmental position in the treatment condition" resulted in "all conspiratorial statements [being] either pro-environmental or environmental-skeptic." I *think* what the authors are trying to say is that they wanted to evaluate categorization by conspiracism in general, not categorization by conspiracism by environmental position. They should clarify on this point (and offer examples.)

3. Finally, I admit I don't fully understand why conspiracy theories are inherently "offensive." Consider those who believe in JFK assassination theories. Given how widely held such beliefs are among U.S. citizens, it's hard to understand how the belief itself is "offensive" in any meaningful way. The authors should either explain this term or use a more precise one.

But again, this is a strong revision. I look forward to reading the published version.

7. PLOS authors have the option to publish the peer review history of their article (what does this mean?). If published, this will include your full peer review and any attached files.

Reviewer #1: No

Reviewer #2: No

---

## [Author Response · Author response to Decision Letter 1]

16 Feb 2022

EDITOR

We have reviewed our reference list and ensured it is complete and correct. We have not cited papers which have been retracted. The only modification made to the reference list is its order (reference 39 became 33, so that references 33 to 38 became references 34 to 39). 

REVIEWER 2

1. If I were the authors, I would indeed report results for all subjects, including those who failed the post treatment attention check. The authors acknowledge that these results are what *should* be reported; the results don't change (they say) if those subjects are included; and, perhaps most importantly, this paper is going to be published, and it would be unfortunate if readers focused on this error, rather than the substantive contribution of the paper. In short, I think it's in their interest, and the long-term interests of this paper, to make this change.

We have made the changes suggested by the reviewer, moving the results with exclusion of inattentive respondents in the supplementary information (in S3, after the description of attention checks) and replacing them in the main text with the results without exclusion (that were reported in S4 in the revision). We agree with the reviewer that it would be unfortunate if readers focused on the possible post-treatment bias when reading the article. Moreover, our conclusions remain unchanged, with only a slight reduction in effect sizes when inattentive respondents are included. We also modified the figures accordingly.

In the Participants section of all studies, we removed all references to attention checks and modified the number of participants included in the analyses. Also, when introducing Study 1, as we deviate from the pre-registration by not excluding inattentive respondents, we added the following paragraph:

“In our pre-registered studies, we planned to exclude participants who failed attention checks. However, because attention checks were implemented post-treatment, these exclusions could bias our causal estimates [33]. Accordingly, we deviate from our pre-registrations and include all respondents in the analyses reported below. In the supporting information (S3), we report pre-registered analyses on attentive respondents yielding identical substantive conclusions.“ (p.11)

2 I would appreciate more details on the modifications made in Study 2. Right now, I don't think I fully grasp how the alignment of "conspiratorial dimension of statements with environmental position in the treatment condition" resulted in "all conspiratorial statements [being] either pro-environmental or environmental-skeptic." I *think* what the authors are trying to say is that they wanted to evaluate categorization by conspiracism in general, not categorization by conspiracism by environmental position. They should clarify on this point (and offer examples.)

We agree with the reviewer on the need to clarify the design of Study 2, its aim and its differences with Study 1. We realized that it may be clearer to describe Study 2 as an experiment eliminating a confound rather than testing an alternative hypothesis. Indeed, we believed that the unexpected results of Study 1 may be due to the fact that people categorize targets according to conspiracism in general, and thus that having only conspiratorial statements in our treatment condition may be a confounding factor blurring categorization by environmental position. We therefore wished to make conspiracism vary in our new design, which is why only half of the statements in the treatment condition are conspiratorial in Study 2. Because we were still mainly focusing on the potential use of conspiratorial sentences to strengthen categorization across another coalitional dimension, we did not create a new conspiracy dimension orthogonal to race and environment. Instead, we aligned the conspiratorial dimension with environmental position such that either all four pro-environmental statements are conspiratorial and no environmental-skeptic statements are, or vice versa. We then tested whether conspiratorial arguments strengthen categorization by environmental position if only one side uses them. We therefore reframed both the discussion of Study 1, the introduction of Study 2 and its conclusion to clarify these points:

“(...) A possible confound influencing the results of this study is that conspiratorial justifications could serve as an indicator of affiliation with an independent coalition composed of all conspiracy theorists (...) ” (p. 13, discussion of Study 1)

“Study 2 was designed to investigate further the unexpected results of Study 1, by eliminating the potential confound that conspiratorial justifications may serve as an indicator of affiliation with an independent coalition composed of all conspiracy theorists. To do so, we modify the treatment condition by eliminating half of the conspiratorial frames compared to Study 1. As our focus remains on the potential use of conspiratorial sentences to boost categorization across another coalitional dimension, we do not create a new conspiracy dimension orthogonal to race and environment. Instead, we align the conspiratorial dimension with environmental position such that either all four pro-environmental statements are conspiratorial and no environmental-skeptic statements are, or vice versa. We then test whether conspiratorial arguments strengthen categorization by environmental position if only one side uses them. If this is true, we expect categorization by environmental position to increase in the treatment group compared to the control group, as all conspiracy theorists now share the same environmental stance. Furthermore, if indeed conspiratorial asymmetries boost environmental position as a coalitional cue, we expect categorization by race to decrease in the treatment group.“ (p.14-15)

“(...) Hence, the findings of Study 2 do not support the prediction that conspiratorial frames boost categorization by environmental position when only one side uses them, as only a weak effect in the expected direction was found.” (p.16)

Finally, we modified the general discussion to clarify the findings of Study 2:

“Study 2 was designed to eliminate a confound that could influence Study 1’s results, namely that conspiratorial justifications may serve as an indicator of affiliation with an independent coalition composed of all conspiracy theorists. However, Study 2 only found a weak effect in favor of the coalitional cue conveyed by conspiracy theories when removing this confound.” (p. 16-17) 

3. Finally, I admit I don't fully understand why conspiracy theories are inherently "offensive." Consider those who believe in JFK assassination theories. Given how widely held such beliefs are among U.S. citizens, it's hard to understand how the belief itself is "offensive" in any meaningful way. The authors should either explain this term or use a more precise one.

We agree with the reviewer that conspiracy theories can take multiple forms and thus sometimes do not appear as explicitly offensive. In the case of JFK assassination conspiracy theories, some groups were accused of the assassination such as the CIA, the Mafia, Lyndon Johnson, Fidel Castro, the KGB, etc. But it is true that sometimes the theory just runs as “JFK was assassinated” and that the reference to malevolent groups is rather implicit. We have reframed and nuanced the part of the introduction where the offensive dimension of conspiracy theories is discussed to reflect more diverse forms of conspiracy theories (including examples): 

“A conspiracy theory is commonly defined as the belief that a group of agents secretly acts together with malevolent intent [18,19]. Most conspiracy theories are thus inherently offensive: they accuse some actors of harming innocent people, either actively (as in the chemtrail conspiracy) or passively by concealing relevant information and “covering up tracks”. Another common case is that conspiracy theories deny grievances or important achievements of certain actors (e.g. Holocaust deniers or the 9/11 Truth Movement; moon-landing hoax), thereby also fostering inter-group conflict. Moreover, many conspiracy beliefs oppose mainstream narratives and are often held by small minorities (e.g. Reptilian conspiracies), thereby also possessing a fringe element.” (p.4)

But again, this is a strong revision. I look forward to reading the published version.

Thank you!

---

## [Editor Report · Decision Letter 2]

28 Feb 2022

Do conspiracy theories efficiently signal coalition membership? An experimental test using the “*Who Said What?*” design

PONE-D-21-29587R2

Dear Dr. Mus,

We’re pleased to inform you that your manuscript has been judged scientifically suitable for publication and will be formally accepted for publication once it meets all outstanding technical requirements.

Kind regards,

Peter Karl Jonason

Academic Editor

PLOS ONE
---

## [Editor Report · Acceptance letter]

2 Mar 2022

PONE-D-21-29587R2 

Do conspiracy theories efficiently signal coalition membership? An experimental test using the “*Who Said What?*” design 

Dear Dr. Mus:

I'm pleased to inform you that your manuscript has been deemed suitable for publication in PLOS ONE. Congratulations! Your manuscript is now with our production department. 

Kind regards, 

on behalf of

Dr. Peter Karl Jonason 

Academic Editor

PLOS ONE